# (PEEP) Predicting Enzyme Functionality with its Molecule Mate – an Attentive Metric Learning Solution

## Abstract

Annotating the functions of proteins (*e.g.*, enzymes) is a fundamental challenge, due to their diverse functionalities and rapidly increased number of protein sequences in databases. Traditional approaches have limited capability and suffer from false positive predictions. Recent machine learning (ML) methods reach satisfactory prediction accuracy but still fail to generalize, especially for less-studied proteins and those with previously uncharacterized functions. To address these pain points, we propose a novel ML algorithm, PEEP, to predict enzyme functionality, which integrates biology priors of protein functionality to regularize the model learning. To be specific, at the input level, PEEP fuses the corresponding molecule into protein embeddings to gain their reaction information; at the model level, a tailored self-attention is leveraged to capture importance residues which we found are aligned with the active site in protein pocket structure; at the objective level, we embed functionality label hierarchy into metric learning objectives by imposing larger distance margin between proteins that have less functionality in common. PEEP is extensively validated on three public benchmarks, achieving up to $4.6\%, 3.1\%, 3.7\%$ improvements on F-1 scores compared to existing methods. Moreover, it demonstrates impressive generalization to unseen protein sequences with unseen functionalities.[1]

## 1 Introduction

Identifying the function of enzymes is a major barrier for the development of biocatalyst for industrial or pharmaceutical applications. High throughput sequencing techniques have enabled millions of sequences of uncharacterized proteins to be added to protein databases everyday (Uni, 2023). Although UniProtKB–a central hub for the collection of functional data on proteins–has grown to over 250M sequences, only around 570K ($\sim 0.3\%$) sequences have been manually annotated Swiss-Prot (Boutet et al., 2007), with computational methods that bridge the sequence-annotation gap. Unfortunately, the critical assessment of protein function annotation (CAFA) study found that $\sim 40\%$ of the computational annotations are incorrect (Radivojac et al., 2013). Additionally, nearly a third of known bacterial proteins lack a characterized homolog to infer function from and their function remains unknown (Price et al., 2018). Thus, designing novel computational tools that can accurately annotate enzymatic function and generalize to novel substrates and new-to-nature reactions is critical for the development of protein-based biotechnologies.

In past few years, significant advancements in machine learning (ML) have revolutionized various biological research domains, such as protein structure (*e.g.*, Jumper et al., 2021; Baek et al., 2021; Watson et al., 2023) and stability (*e.g.*, Diaz et al., 2023; Chen et al., 2022; Umerenkov et al., 2022), single-cell RNA-sequencing (Weinberger et al., 2023), population genetics (Schrider & Kern, 2018), drug discovery (Zhang et al., 2023), and many others. As for enzyme function prediction, several ML frameworks have been recently presented that formulate the task as a classification problem (*e.g.*, Ryu et al., 2019; Sanderson et al., 2023; Dalkiran et al., 2018). Here, the community has leveraged the enzyme commission (EC) number of annotated enzymes, which is a classification ontology for the chemical reactions catalyzed by enzymes (Webb et al., 1992), to build classical ML pipelines.

---

[1]Codes are included in the supplement.

However, these frameworks have limited application and generalization due to the limited number of established EC numbers in their training sets compared to the actual reaction space of enzymes and/or the imbalance of annotated sequences between available EC numbers, respectively.

A recent study by Yu et al. (2023) proposed the CLEAN (contrastive learning-enabled enzyme annotation) framework: a retrieval-based framework that uses a contrastive lose for metric learning. CLEAN uses the pairwise distance between a query sequence and each EC cluster (the average embedding of all sequences with the same EC number) to retrieve the EC number(s). This approach significantly outperformed classification deep learning frameworks, such as ProteInfer (Sanderson et al., 2023), DeepEC (Ryu et al., 2019), and DEEPre (Li et al., 2018), on two independent test sets. Furthermore, CLEAN's performance was most impressive on EC numbers with less than ten representative sequences, highlighting the effectiveness of contrastive learning compared to multi-label classification for enzyme function prediction. However, the CLEAN framework employs the triplet loss to distinguish proteins between enzymes at the substrate class level (4th EC number). This design decision fails to leverage the hierarchical nature of enzyme function. Furthermore, the framework is not designed to generalize to proteins with novel functionality or substrate scope, which, by definition, will lack an EC number. Another crucial drawback in existing approaches (*e.g.*, CLEAN) is the neglect of biological priors such as knowledge about related chemical reactions or protein properties. Directly plugging machine learning algorithms can work but is limited in generalization (*e.g.*, Yu et al., 2023; Sanderson et al., 2023; Ryu et al., 2019).

To address the aforementioned limitations, we proposed a metric learning framework, *i.e.*, PEEP, that incorporates biological priors in order to learn richer representations, shorten training time, and improve generalization beyond established EC numbers. In detail, PEEP considers biological priors in multiple facets: at the input level, we fuse the SMILES representation of a protein's cognate ligands as an alternative route for providing substrate-scope information; at the model level, we add a transformer layer to facilitate learning functional residues associated with an enzyme function, such as residues that participate in catalysis or ligand binding; at the objective level, we propose a metric learning objective that captures the hierarchical nature of EC numbers to appropriately weight dissimilarity at each EC level. Our contributions are summarized below:

- (Algorithm) We propose a novel metric learning based framework, PEEP, to identify enzyme with desired function.

- (Algorithm) Our PEEP incorporates biology-aware designs to regularize the learning of protein functionality: (1) integrating the cognate ligands' embeddings as a complementary source of features; (2) inserting an attentive module in order to capture key residues for protein functionality; (3) introducing an EC-aware training objective to enhance the metric learning capability and capture the hierarchical characteristics of EC annotations.

- (Data-Engineering) We filter publicly available training sets by taking into account sequence similarity (10% and 30% thresholds) in order to mitigate over-fitting to overly represented regions of sequence space.

- (Experiments) On two public benchmarks, we empirically demonstrate that our framework surpasses all existing methods by clear margins and reaches state-of-the-art performance, especially when sequence clustering is considered. Specifically, on Price and New, when controlling sequence similarity in the training set (30% threshold), we outperform the most competitive baseline by $4.6\%$ and $1.1\%$ in terms of the F-1 score, respectively.

- (Extra Applications) Furthermore, PEEP enables a series of applications spontaneously, such as protein-ligand binding prediction and active site prediction.

## 2 RELATED WORKS

**Enzyme Function Prediction.** The protein community has been utilizing computational tools to infer protein functions for a long time and has developed various methods based on sequence similarity (*e.g.*, Altschul et al., 1990; Desai et al., 2011; Altschul et al., 1997), protein homology (Krogh'f & Brown, 1994; Steinegger et al., 2019), protein structures (Zhang et al., 2017) and sequence motif (Bairoch, 1991). ML-based algorithms have been proposed as strong competitors to tackle this challenge. Existing deep learning frameworks works mostly use the multi-label classification framework, such as Proteinfer (Sanderson et al., 2023), DeepEC (Ryu et al., 2019), and DEEPre (Li et al.,

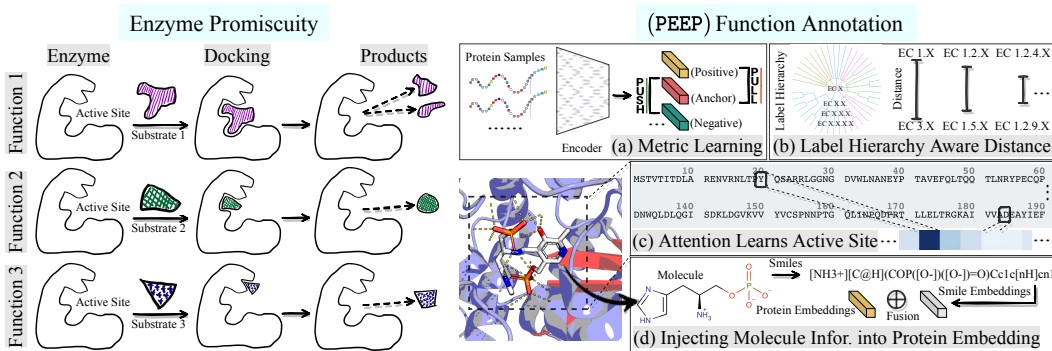

Figure 1: The overview of our proposed PEEP framework. (*Left*) PEEP aims to predict the enzyme functionality, by annotating its functionality. Enzyme functions usually are defined by the associated chemical reactions. (*Right*) (*a*) PEEP is a metric learning based approach. (*b*) It learns enzyme embeddings with respect to their EC label hierarchy. (*c*) A tailored attention mechanism is leveraged to capture important residues, *e.g.*, these ones at the active site. (*d*) Associated molecule knowledge is fused into protein embeddings through a customized neural network.

2018), which suffers from imbalance training datasets often observed in biology. Recently, new paradigms have been introduced to tackle this challenge. Xu & Wang (2022) redefines the protein function prediction problem as a machine translation problem that aims at translating the description of a function to the amino acid sequence with the goal of identifying novel gene ontologies (GO terms), and has been extended (Xu et al., 2023) to a multilingual translation framework that can enable diverse applications including protein function prediction. Additionally, Yu et al. (2023) proposes a metric learning framework to maximize the distance of protein embeddings between different functions, and minimize those between the same functions, achieving state-of-the-art (SoTA) performance. However, they only adopt a simple triplet loss to contrast the samples without considering the hierarchical nature of EC numbers and do not incorporate any biological priors to improve generalization functions lacking an established EC number.

**Protein Representation Learning.** Prior research endeavors have sought to learn the representation of proteins by exploiting diverse protein modalities. Early sequence-based methods define rules to extract physiochemical or statistical features from protein sequences (*e.g.*, Klein et al., 1985; 1986; Feng & Zhang, 2000; Wang et al., 2017). Later on, machine learning methods that utilize word2vec, doc2vec (*e.g.*, Asgari & Mofrad, 2015; Kimothi et al., 2016; Ng, 2017) or recurrent neural networks (Mazzaferro, 2017; Alley et al., 2019) are proposed. The development of transformers subsequently increases the model capacity, and protein language models are becoming increasingly popular (*e.g.*, Elnaggar et al., 2020; Rao et al., 2019; Rives et al., 2021; Lin et al., 2022). Furthermore, Multiple-sequence alignment (MSA) models (*e.g.*, Rao et al., 2021; Biswas et al., 2021; Meier et al., 2021) learn to represent a protein in its evolutionary context. Structure-based models aim to encode spatial information in protein structures into their representation by using 3D convolutional neural networks (CNN) (*e.g.*, Derevyanko et al., 2018; Shroff et al., 2020; Lu et al., 2022; d'Oelsnitz et al., 2023), graph neural networks (GNN) (*e.g.*, Gligorijević et al., 2021; Hermosilla & Ropinski, 2022; Zhang et al., 2022), or transformers (*e.g.*, Diaz et al., 2023; Rives et al., 2021; Lin et al., 2022). While these methods achieve superior performance on some tasks, for most tasks the sequence-based models still dominate the performance thanks to the availability of the massive amounts of protein sequences (Xu et al., 2022). Therefore, this paper uses the sequence embeddings of ESM2 (Lin et al., 2022) for proteins and uses the Momuntum Contrast (MoCo) technique (He et al., 2020) as a backbone to further improve the representations at the fine-tuning phase.

## 3 METHODOLOGY

**Overview of PEEP.** PEEP is a metric learning framework designed to learn an embedding space where the distances between protein sequences reflect the similarity between their functionalities. For a protein sequence $x$ with $n$ amino acids and its functional annotation $y$ in the training set, we sample another sequence $x'$ who shares the same functional annotation $y$. Subsequently, we compute the representations of these proteins using the protein sequence encoder $\mathcal{E}$, denoted as $h$

and $h'$. More details on the structure of $\mathcal{E}$ and the methods to obtain $h$ and $h'$ are deferred to Section 3.2 and 3.3.

The base pipeline of PEEP adopts a similar design to MoCo (He et al., 2020). It passes protein embeddings $h$ and $h'$ from the pre-trained encoder $\mathcal{E}$ through two multi-layer perceptrons (MLPs), denoted as $\mathcal{M}$ and $\mathcal{M}'$, producing two final latent representations $z = \mathcal{M}(h)$ and $z' = \mathcal{M}'(h')$. At the training phase, PEEP maintains a queue of the $Q$ most recent $z'$, maximizing the cosine similarity between $z$ and $z'$ and meanwhile minimizing the similarity between $z$ and the representations stored in the queue. The training objective for optimizing $\mathcal{M}$ and $\mathcal{M}'$ in PEEP defined as:

$$\mathcal{L}(z, z') = -\log \frac{\exp(s(z, z')/\tau)}{\exp(s(z, z')/\tau) + \sum_{i=1}^{K} \exp(s(z, z'_{q,i})/\tau)}, \tag{1}$$

where $s(\cdot, \cdot)$ denotes the cosine similarity function and $z'_{q,i}$ signifies the $i$-th entry in the queue of stored representations which are derived based on $\mathcal{M}'$. The $\mathcal{M}'$ is implemented as a momentum-based moving average of $\mathcal{M}$, a strategy proposed to ensure consistency (He et al., 2020).

At the inference stage, for an unknown protein sequence $x_u$, PEEP first extracts its latent representation using the aforementioned pipeline, denoted as $z_u$. Next, for every functional annotation presented in the training dataset, PEEP gathers all protein sequences associated with it and computes their average latent embedding, which is then designated as the function's representation. Finally, to predict the functionality of $x_u$, PEEP calculates the distance between $z_u$ and each function's representation and ranks the functional annotations accordingly. If a protein has multiple function annotations, the max separation algorithm (Yu et al., 2023) will be utilized (refer to Appendix A1) to decide which set of annotations should be selected for the prediction. Detailed designs of PEEP at the objective, model, and input levels are depicted below, respectively.

## 3.1 INTEGRATING ENZYME LABEL HIERARCHY INTO METRIC LEARNING

In this section, we describe our biology-ware innovations at the objective level of PEEP. Protein functional annotations, such as EC number or GO terms, have hierarchical structures. Specifically, the EC number associates a protein sequence with a chemical reaction and lies in four-digit hierarchical classes. From left to right, the digits correspond to the reaction class, subclass, and sub-subclass, and a serial number that is substrate-specific. As an illustration, considering the functional annotation of tripeptide aminopeptidase (*i.e.*, "EC 3.4.11.4"), its components indicate the following:

⋆ EC **3**: the hydrolase class, which uses water to break a covalent bond;

⋆ EC 3.**4**: the subclass of hydrolases that cleaves peptide bonds;

⋆ EC 3.4.**11**: the sub-subclass that cleaves the N-terminus of a polypeptide;

⋆ EC 3.4.11.**4**: the substrate is a tripeptide.

Intuitively, the representation distance between two different proteins is expected to decrease as their function annotations diverge only at the latter digits, owing to their high similarity in chemical reactions. For instance, in contrast to proteins with EC 3.4.11.4, those with EC 3.4.11.1 (*i.e.*, leucyl aminopeptidase) merely exhibit differences in their substrates, while proteins with EC 3.4.21. - are serine endopeptidases that have a relatively lower similarity. This hierarchical difference can be advantageous for learning biologically meaningful distances among different proteins. It is a potential benefit that current methods neglect. In our case (Figure 1b), PEEP uses an additional ontology-aware objective to penalize cosine similarity between different proteins, aiming to acquire knowledge of this label hierarchy. The detailed calculation is described as follows.

For two protein sequences $x_i$ and $x_j$ accompanied by their functional annotations $y_i$ and $y_j$, we compute their representations denoted as $z_i$ and $z_j$ and derive their similarity score as $s(z_i, z_j)$. Subsequently, a function diff is introduced to quantify the dissimilarity between their functional annotations $y_i$ and $y_j$ along with the label hierarchy. The minimization objective is then induced as $\mathcal{L}_{i,j} = \max\{\text{diff}(y_i, y_j) \cdot s(z_i, z_j) - 1, 0\}$, excepting that our goal is decreasing the cosine similarity when the difference is large. In Section 4.3, we provide the comparison with using the reciprocal version of $\mathcal{L}_{i,j}$ (*i.e.*, $\mathcal{L}'_{i,j} = \max\{1 - 1/(\text{diff}(y_i, y_j) \cdot s(z_i, z_j)), 0\}$). If we consider a batch of sequences $\mathcal{B}$, PEEP juxtaposes each sequence with every other in a pairwise fashion and accumulate the corresponding objective values as $\mathcal{L} = \frac{1}{|\mathcal{B}|^2} \sum_{1 \leq i,j \leq |\mathcal{B}|} \mathcal{L}_{i,j}$.

## 3.2 Capturing Crucial Residues From Active Sites via Tailored Self-Attention

In this section, we describe our biology-ware innovations at the model level of `PEEP`. Let us recall the `CLEAN` framework (Yu et al., 2023) that treats all amino acids to be equally important and computes the straightforward average of representations from each amino acid to establish the representation of a whole protein sequence. However, all amino acids (or residues) play distinctive roles in determining a protein's functionality, and the ones that lie in the active site (a region that interacts with specific molecules) contribute significantly (Dassi et al., 2021). Therefore, an ideal solution should emphasize its attention on these crucial residues. To meet the goal, we deploy a self-attention mechanism (Figure 1, c) to model the residue importance within protein sequences.

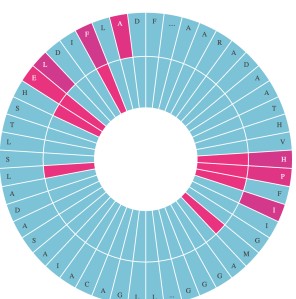

To be specific, our protein sequence encoder $\mathcal{E}$ comprises two main components: a pre-trained protein feature extractor denoted as $f(\cdot)$ such as ESM (Rives et al., 2021; Lin et al., 2022), and two projection weights, $\mathbf{W}_1$ and $\mathbf{W}_2$, employed within the attention mechanism. We input $\boldsymbol{x}$ into $f(\cdot)$ and then multiply it with $\mathbf{W}_1$ and $\mathbf{W}_2$, respectively, with the formulas $\boldsymbol{q} = f(\boldsymbol{x})\mathbf{W}_1$ and $\boldsymbol{k} = f(\boldsymbol{x})\mathbf{W}_2$. It then adheres to standard procedures for deriving the probability score matrix by taking the product of $\boldsymbol{q}$ and $\boldsymbol{k}$, and performing normalization and `softmax`. The resulting matrix is $\mathcal{P} = \texttt{softmax}(\boldsymbol{q}\boldsymbol{k}^\top/\sqrt{d})$, with $d$ denoting the output dimension of $f(\cdot)$.

Another biological characteristic of enzymes is that *active sites are sparse*. Thus, we introduce a method variant $\mathcal{P}_{\text{sparse}}$ that only keeps the top $K$ largest entries in each row as non-zero to encourage sparsity. Finally, $\mathcal{P}$ (or $\mathcal{P}_{\text{sparse}}$) is multiplied by $f(\boldsymbol{x})$ and we further average over all amino acids to obtain the representation $\boldsymbol{h}_s$ for a whole protein sequence. Appendix A1 offers a comprehensive exposition of different attention designs. Visualizations as shown in Figure 2 demonstrate a strong correlation between residues at the active sites and are emphasized by our attention, suggesting the effectiveness of our proposal.

Figure 2: The visualization of a protein sequence with its active sites highlighted and the learned attention scores at each amino acid. It evidences that our attention is capable of capturing the underlying structural property of enzymes.

## 3.3 Fusing Molecule Information into Enzyme Representations

In this section, we present our biology-ware innovations at the input level of `PEEP`. It fuses the information of a protein's cognate ligands with its sequence representation $\boldsymbol{h}_s$, to inject substrate-scope knowledge, as presented in Figure 1 ($d$). Given an enzyme, `PEEP` traverses all the associated function labels, retrieving the most intricate substrate and the corresponding resulting product involved in the reaction. This information is sourced from publicly accessible datasets[2]. Subsequently, we employ another pre-trained feature extractor (Chithrananda et al., 2020) to extract their respective representations (*a.k.a.* SMILES embeddings), denoted as $\boldsymbol{h}_{l,1}, \ldots, \boldsymbol{h}_{l,n_l}$.

At each training step, `PEEP` randomly samples one from the ligands' SMILE embedding and integrates it with the protein's sequence representation. Our results in Section 4.3 indicate that it is an effective sampling strategy to pick ligands. The detailed fusion methods can be either (①) learning a coefficient for weighted averaging or (②) passing through a trainable MLP.

① $\boldsymbol{h} = \boldsymbol{p} \odot \boldsymbol{h}_{l,i} + (1-\boldsymbol{p}) \odot \boldsymbol{h}_s$, where $1 \leq i \leq n_l$ is a random index and $\boldsymbol{p}$ is the learnable coefficient,

② $\boldsymbol{h} = \mathcal{M}_f([\boldsymbol{h}_i; \boldsymbol{h}_s])$, where $\mathcal{M}_f$ is a trainable MLP.

Moreover, to augment the learning process, `PEEP` uses an auxiliary classifier to distinguish between "positive" and "negative" fused representations $\boldsymbol{h}_s$. They are obtained by integrating associated and non-associated pairs of protein sequences and ligands, respectively.

In the inference phase, two modes are established for the handling of previously unencountered proteins. The first mode gathers all sequences linked to each functional annotation within the training dataset, along with their corresponding ligands. For a given annotation, `PEEP` combines the representations of each sequence with those of the corresponding individual ligands, computing the average of all conceivable pairings to form the representation of the chosen functional annotation.

---

[2]https://www.rhea-db.org/help/search-ec-number

If it needs to handle the protein with an unknown functionality, `PEEP` will compute the distance between its representation and that of each function. In the second mode, we simplify the process by treating all ligand representations as zero vectors when computing the representations of functional annotations and the given protein sequence. This approach provides a straightforward solution for handling unobserved protein sequences. The comparison of these two modes is in Appendix A3.2.

## 4 EXPERIMENTS

### 4.1 IMPLEMENTATION CONFIGURATIONS

**Datasets.** The protein sequences in our training set are sampled from Swiss-Prot (Boutet et al., 2007). It is a subset of the comprehensive UniProt dataset, which is meticulously reviewed by humans. This ensures that our model learns from human-curated annotations. Adopting the data filtering methodology from Yu et al. (2023), we obtain a training set with around 220K samples. Subsequently, we cluster and subsample the data using MMSeqs2 using various sequence identity cutoffs at 10% and 30%, which helps us to remove the homologs. We evaluate `PEEP` of EC annotation performance on three established benchmarks. The established datasets are (1) `Price-149` (or `Price`), which was assembled by Sanderson et al. (2023) as a challenging dataset since the sequences inside this dataset have been incorrectly or inconsistently labeled in renowned databases such as the Kyoto Encyclopedia of Genes and Genomes (KEGG); (2) `New-392` (or `New`), which consists of 392 enzymes sequences distributed across 177 distinct EC numbers. (3) `CATH` (Sillitoe et al., 2021), which classifies protein sequences into multiple (structural) domains with functional labels available. We collect proteins from the superfamily of 3.40.710.10 and use them to assess the models. The statistics of these benchmarks and the details of data sampling are summarized in Table A7 and Section A2.5. The results are available in Section A3.

**Network Architecture.** We leverage both ESM-1b (Rives et al., 2021) and ESM-2 (Lin et al., 2022) as the pre-trained protein feature extractor $f$ to obtain the sequence representations of proteins. The architectural details of other components in `PEEP`, including the MLP and the attention module's weight shapes, are available in Appendix A2.2.

**Extracting the Ligands Embeddings.** We obtain the SMILES notations of protein's cognate ligands from the Rhea dataset (Bansal et al., 2022), which is an expert-curated database of chemical and transport reactions of characterized enzymes. To extract the representations of SMILES notations, we use ChemBERTa (Chithrananda et al., 2020), a language model pretrained on a chemical dataset called PubChem (Kim et al., 2019). In our experiments, we explore two variants of the extracted embeddings of SMILES notations: (1) using the averaged representation without the special tokens and (2) using only the representation of the `[CLS]` token as the representation. In Section 4.3, comparisons of both variants show that exploiting `[CLS]` tokens leads to better performance.

**Baselines.** We follow Sanderson et al. (2023) to implement an alignment-based baseline. Specifically, we utilized BLASTp to discern proteins in the training set that bear high similarity to an unknown test protein. The functions of these identified proteins are then assigned as the predicted functional annotation for the test protein. Moreover, several baseline algorithms are involved into comparisons: (1) classification methods: DeepEC (Ryu et al., 2019), DEEPre (Li et al., 2018), ECPred (Dalkiran et al., 2018), ProteInfer (Sanderson et al., 2023); (2) retrieval methods: CLEAN (Yu et al., 2023) and (3) a translation-based method, known as Bio-Translator (Xu et al., 2023), which is noteworthy for its zero-shot capability for multiple applications. More details of utilizing Bio-Translator to perform multi-label classification is presented in Section A2.

### 4.2 `PEEP` ENABLES SUPERIOR ENZYME FUNCTIONALITY PREDICTION

**Comparison between `PEEP` and Existing SoTAs.** Firstly, we conduct experiments using the training set with a sequence identity of 10%, and report the performance of `PEEP` in Table 1 with other baselines, *i.e.,* ProteInfer and CLEAN. Following Yu et al. (2023), we compare the performance of `PEEP` in terms of the recall, precision, and the F-1 score with the baseline methods. It can be clearly seen from the table that our method reaches the highest performance among all methods. Compared with the most competitive baseline CLEAN, our method achieves an improvement of $\{3 \sim 3.8\%, 2.6 \sim 7.8\%, 3.1 \sim 3.7\%\}$ in terms of the {recall, precision, F-1} scores on `Price` and `New` using the ESM-2 35M as the pre-trained feature extractor. A similar improvement is observed when switching to the larger ESM-1b 650M, where `PEEP` outperforms CLEAN consistently on the

Table 1: Quantitative comparison of `PEEP` with the state-of-the-art EC number prediction tools with training sets at different levels of identity cut-off.

| Training Sets | Method | Price | | | New | | |
|---|---|---|---|---|---|---|---|
| | | Rec. | Prec. | F-1 | Rec. | Prec. | F-1 |
| 10% | ProteInfer | 0.046 | 0.095 | 0.059 | 0.111 | 0.178 | 0.115 |
| | Bio-Translator | 0.795 | 0.025 | 0.049 | 0.773 | 0.018 | 0.035 |
| | CLEAN (ESM-1b 650M) | 0.230 | 0.299 | 0.235 | 0.314 | 0.335 | 0.312 |
| | CLEAN (ESM-2 35M) | 0.187 | 0.261 | 0.204 | 0.241 | 0.326 | 0.256 |
| | PEEP (ESM-1b 650M) | 0.232 | 0.315 | 0.241 | 0.308 | 0.382 | 0.325 |
| | PEEP (ESM-2 35M) | 0.217 | 0.339 | 0.241 | 0.279 | 0.352 | 0.287 |
| 30% | ProteInfer | 0.086 | 0.197 | 0.105 | 0.129 | 0.210 | 0.140 |
| | CLEAN (ESM-1b 650M) | 0.257 | 0.336 | 0.267 | 0.412 | 0.388 | 0.362 |
| | CLEAN (ESM-2 35M) | 0.217 | 0.308 | 0.235 | 0.338 | 0.360 | 0.316 |
| | PEEP (ESM-1b 650M) | 0.237 | 0.391 | 0.269 | 0.376 | 0.432 | 0.374 |
| | PEEP (ESM-2 35M) | 0.260 | 0.393 | 0.281 | 0.320 | 0.401 | 0.327 |
| 100% | BLASTp | 0.375 | 0.508 | 0.385 | - | - | - |
| | DeepEC | 0.072 | 0.118 | 0.085 | 0.217 | 0.298 | 0.230 |
| | ECPred | 0.020 | 0.020 | 0.020 | 0.095 | 0.118 | 0.100 |
| | ProteInfer | 0.138 | 0.243 | 0.166 | 0.284 | 0.409 | 0.309 |
| | CLEAN (ESM-2 35M) | 0.430 | 0.541 | 0.456 | 0.441 | 0.563 | 0.459 |
| | PEEP (ESM-2 35M) | 0.412 | 0.572 | 0.458 | 0.426 | 0.652 | 0.474 |

precision and F-1 score by $1.6 \sim 4.7\%$ and $0.6 \sim 1.3\%$, respectively. Then, we continue to conduct experiments on the training set with $30\%$ sequence identity, where it demonstrates that our framework preserves its advantage against all baseline methods. Specifically, `PEEP` surpasses CLEAN by $4.1 \sim 8.5\%$ and $0.2 \sim 4.6\%$ in terms of the precision and the F-1 scores respectively, while maintaining the same level of recall scores. This indicates the effectiveness of our algorithm under a slightly data-richer setting. Finally, we provide the recall, precision, and F-1 score of `PEEP` experimented with the full data (*i.e.*, training set with $100\%$ sequence identity). The table shows that our method achieves the highest performance among all the methods. This series of experiments verifies the superior generalization ability of `PEEP`.

**`PEEP` Can Generalize to Unseen Protein with Unseen Functionalities.** In previous sections we assess `PEEP`'s performance on assigning unseen proteins to seen functionalities. Subsequently, we assess its performance on proteins linked to functions that have not been encountered during training. It is essential to emphasize that these proteins also remain unseen during training. Our objective is to explore whether the representations of these unfamiliar proteins can create distinct clusters compared to proteins with different functionalities. In particular, we train `PEEP` on the training dataset with

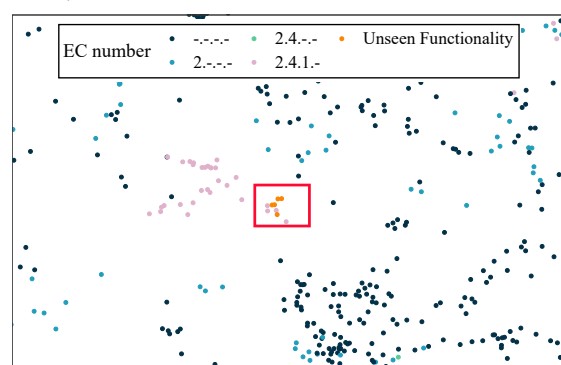

Figure 3: UMAP of protein representations associated with different EC numbers and the unseen functionality.

$10\%$ sequence identity and extract protein representations associated with an EC number of 2.4.1.17, representing previously unseen functionalities within the training dataset. We extract and visualize the protein representations with UMAP in Figure 3, from which we can observe that (1) the proteins from unseen functionalities exhibit close clustering despite their diverse protein sequences (see Appendix A2.4), suggesting that `PEEP` demonstrates an awareness of functional homogeneity; (2) the distances between protein representations from the unseen functionality and those with EC numbers starting with "2.4.1" (or having EC 2.4.1.-) tend to be shorter than distances to other protein, suggesting functional similarity with proteins associated with EC 2.4.1.-. These observations demonstrate that the utilization of "PEEP" can effectively address previously unencountered functionalities.

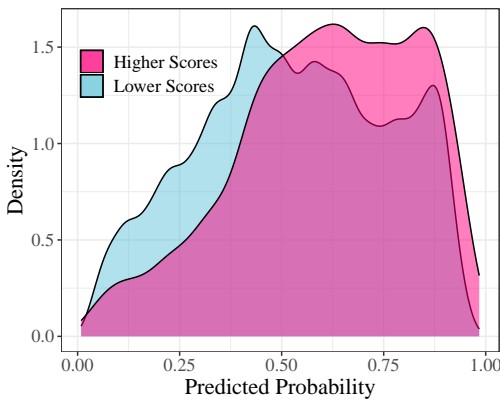

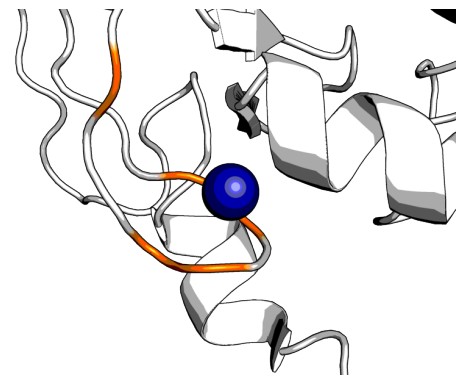

Figure 5: Visualization of the structure of the selected protein with the PDB ID of 5CAW. The highlighted residues marked in orange color indicate those assigned with greater attention scores by PEEP. The blue ball is the ligand.

Figure 4: The distribution of the predicted probability of residues being in active sites, categorized by attentive scores.

### 4.3 EXTRA INVESTIGATIONS AND ABLATION STUDY

In this section, we demonstrate the versatility of PEEP by spontaneously solving additional tasks, such as active site detection and predicting protein-ligand binding (refer to Appendix A3.1). We investigate various design alternatives in ablation studies. More analyses are in Appendix A3.2.

**PEEP for Zero-shot Active Site Detection.** The attention score matrix $P$ (or $P_{\text{sparse}}$), quantifies the importance of residues and has the potential to identify active sites, albeit through unsupervised means. We conduct experiments to quantitatively evaluate the correlation between attention scores and potential active sites. Due to the inconsistent availability of the precise active site locations, we employ a computational tool called PocketMiner (Meller et al., 2023) as a proxy for estimating the likelihood of a specific residue's presence in crucial regions across all proteins within the New dataset. For PEEP, we average $\mathcal{P}_{\text{sparse}}$ in a column-wise manner as the estimated importance of residues. The Spearman correlation between the prediction of PocketMiner and PEEP is approximately 0.21, indicating a medium strength level of correlation. Moreover, we select the residues with the highest scores in each row of the attention score matrix, following the methodology outlined by Teukam et al. (2023), for the purpose of evaluating the attention score matrix's quality. Figure 4 shows that the residues assigned with higher attention scores have a more right-skewed distribution of predicted probability scores compared to those with lower attention scores, indicating an agreement between PEEP and PocketMiner.

Finally, we initiate the validation process for a single protein (UniProt ID: E0VIU9, PDB ID: 5CAW) with molecules information publicly available. Figure 5 displays the positions of ligands and the crucial residues indicated by attention scores obtained from PEEP. It is evident that the chosen residues are notably situated around the small molecule indicated by the blue sphere, aligning with the protein structure, thus highlighting PEEP's capability to approximately identify active sites.

**Ablation Studies on Techniques.** Ablation studies are conducted on multiple techniques, encompassing the integration of ligand embeddings, attentive modules, and the EC-aware training objective, in order to assess their efficacy. Table 2 presents the performance for all combinations of techniques. Each introduced technique demonstrably enhances performance, underscoring the efficacy of integrating biological priors within our framework.

**Different Attention Designs.** Table 3 presents precision, recall, and F-1 scores obtained from two distinct probability score matrices: $\mathcal{P}$ and $\mathcal{P}_{\text{sparse}}$. Additionally, we compare these results to those of the baseline model, referred to as "No Attention." The table clearly demonstrates that using $\mathcal{P}_{\text{sparse}}$ yields superior performance. This observation aligns with the notion that active sites exhibit sparsity.

**Different Molecule Information Fusion.** In Table 4 we present the precision, recall, and F-1 scores with different methods to fuse the molecule representations. We compare against several variants: ① *Random Fusion*, which randomly selects a cognate ligand to fuse with the protein representation; ② *Negative Fusion*, which combines the representation of a ligand that does not bind with

Table 2: Quantitative comparison of `PEEP` with different techniques enabled. The columns of "Att.", "EC-aware Obj.", "SMILES Emb." indicate the usage of attentive modules, EC-aware training objective and the integration of ligand embeddings, respectively.

| | Techniques | | Price | | | New | | |
|---|---|---|---|---|---|---|---|---|
| Att. | EC-aware Obj. | SMILES Emb. | Rec. | Prec. | F-1 | Rec. | Prec. | F-1 |
| ✗ | ✗ | ✗ | 0.186 | 0.271 | 0.202 | 0.255 | 0.304 | 0.265 |
| ✓ | ✗ | ✗ | 0.185 | 0.289 | 0.212 | 0.279 | 0.315 | 0.275 |
| ✓ | ✓ | ✗ | 0.191 | 0.302 | 0.216 | 0.292 | 0.338 | 0.290 |
| ✓ | ✗ | ✓ | 0.211 | 0.319 | 0.240 | 0.266 | 0.393 | 0.291 |
| ✓ | ✓ | ✓ | 0.217 | 0.339 | 0.241 | 0.279 | 0.352 | 0.287 |

Table 3: Ablation studies on the `Price` dataset for different designs of attentive modules in our proposed `PEEP`.

| Attention | Rec. | Prec. | F-1 |
|---|---|---|---|
| No Attention | 0.186 | 0.271 | 0.202 |
| $\mathcal{P}$ | 0.187 | 0.285 | 0.209 |
| $\mathcal{P}_{\text{sparse}}$ | 0.185 | 0.289 | 0.212 |

Table 4: Ablation studies on the `Price` dataset for techniques to extract and fuse the ligands' embeddings into those of proteins.

| Design | Rec. | Prec. | F-1 |
|---|---|---|---|
| No Fusion | 0.186 | 0.271 | 0.202 |
| Vanilla | 0.171 | 0.266 | 0.190 |
| Random Fusion | 0.191 | 0.325 | 0.225 |
| Negative Fusion | 0.165 | 0.372 | 0.208 |
| Random & Negative Fusion | 0.211 | 0.319 | 0.240 |

Table 5: Ablation studies for different fusion designs in our proposed `PEEP` on the `Price` dataset.

| Design | | Rec. | Prec. | F-1 |
|---|---|---|---|---|
| Weighted Average | Average | 0.204 | 0.303 | 0.226 |
| MLP | Average | 0.197 | 0.389 | 0.230 |
| Weighted Average | [CLS] | 0.211 | 0.319 | 0.240 |
| MLP | [CLS] | 0.165 | 0.355 | 0.210 |

Table 6: Performance comparison of different variants of EC-aware regularization loss on the `Price` dataset.

| Variants | Rec. | Prec. | F-1 |
|---|---|---|---|
| No Reg | 0.186 | 0.271 | 0.202 |
| Multiplication | 0.191 | 0.302 | 0.216 |
| Reciprocal | 0.193 | 0.299 | 0.214 |

the given protein and uses a classifier for distinction; ③ *Random & Negative Fusion*, which employs both the aforementioned techniques; ④ *Vanilla*, which fuses the representations of *all* the cognate ligands with the protein representation. The experimental results demonstrate the effectiveness of the introduced techniques, as each one substantially improves the prediction scores.

Subsequently, we explore different approaches to extract and fuse the representations of ligands and present the results in Table 5. We compare two approaches for obtaining ligand representations: ① the [CLS] token representation extracted by ChemBERTa and ② *Average* representation, which averages all residues. Moreover, we compare two methods for fusion: (1) *Weighted Average*, which learns coefficients for weighted averaging, and (2) *MLP*, designed specifically for fusion. The results indicate that both fusion methods successfully combine ligand representations; nevertheless, learning coefficients to fuse the [CLS] token representation results in superior performance.

**Different Regularization Designs.** We compare different variants of EC-aware regularization: ① the "*multiplication*" version expressed as $\mathcal{L}_{i,j} = \max\{\text{diff}(y_i, y_j) \cdot s(z_i, z_j) - 1, 0\}$; ② the "*reciprocal*" version, as detailed in Section 3.1, expressed as $\mathcal{L}'_{i,j} = \max\{1 - 1/(\text{diff}(y_i, y_j) \cdot s(z_i, z_j)), 0\}$. The performance evaluated on the `Price` benchmark is shown in Table 6, where it shows that the two variants yield comparable performance.

## 5 CONCLUSION

Predicting enzyme functionality is a primary barrier in biomanufacturing. This paper proposes a novel metric learning framework, *i.e.*, `PEEP`. We conduct pioneering efforts to incorporate biology priors like integrating cognate ligands' embeddings, attentive learning of active sites, and considering functionality label hierarchy. Extensive results on multiple public benchmarks show the superiority of `PEEP`. Future works will focus on the wet lab validation of identifying protein generalists.

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

## A1    MORE TECHNIQUE DETAILS

### A1.1    ATTENTION DESIGNS.

As aforementioned, we set up two variants of the calculated probability score matrix, namely, $\mathcal{P}$ and $\mathcal{P}_{\text{sparse}}$. After obtaining these matrices, we calculate the sequence representations blow:

$$\boldsymbol{h} = (\frac{1}{n}\boldsymbol{1}^{\top})\mathcal{P}f(\boldsymbol{x}) \text{ or } \boldsymbol{h} = (\frac{1}{n}\boldsymbol{1}^{\top})\mathcal{P}_{\text{sparse}}f(\boldsymbol{x}),$$

where $n$ indicates the length of the sequence and $\boldsymbol{1}$ is 1-vector with the length of $n$.

## A2    MORE IMPLEMENTATION DETAILS

### A2.1    DATASET STATISTICS

In Table A7 and Table A8 we summarize the statistics of the training sets with different sequence identity and the statistics of the testing sets, respectively.

Table A7: The statistics of training datasets with different levels of sequence identity.

| Sequence Identity | EC number | Number of Proteins |
|---|---|---|
| 10% | 1.-.-.- | 1151 |
| | 2.-.-.- | 3290 |
| | 3.-.-.- | 2516 |
| | 4.-.-.- | 603 |
| | 5.-.-.- | 353 |
| | 6.-.-.- | 259 |
| | 7.-.-.- | 187 |
| 30% | 1.-.-.- | 1521 |
| | 2.-.-.- | 4109 |
| | 3.-.-.- | 3069 |
| | 4.-.-.- | 760 |
| | 5.-.-.- | 461 |
| | 6.-.-.- | 415 |
| | 7.-.-.- | 213 |
| 100% (Full) | 1.-.-.- | 30370 |
| | 2.-.-.- | 88717 |
| | 3.-.-.- | 46075 |
| | 4.-.-.- | 25477 |
| | 5.-.-.- | 15477 |
| | 6.-.-.- | 26574 |
| | 7.-.-.- | 8335 |

### A2.2    HYPER-PARAMETERS

Two different sets of hyperparameters are used for different pretrained feature extractors. For ESM-2 35M which has an output dimension of $480$, we set the output dimension of $\mathbf{W}_1$ and $\mathbf{W}_2$ (*i.e.*, $d$) to be $64$. For ESM-1b 650M, we set the output dimension of the attentive modules to be $256$. The MLPs (*i.e.,* $\mathcal{M}$ and $\mathcal{M}'$) have three linear layers with the shapes of $(d_f, 256), (256, 256), (256, 128)$, respectively, where $d_f$ represents the output dimension of the pretrained feature extractors. When training the networks, we use the AdamW optimizer (Loshchilov & Hutter, 2017) to optimize the parameters for 1000 epochs with a learning rate of $5 \times 10^{-4}$ and a cosine learning rate scheduler.

Table A8: Statistics of testing datasets: `Price`, `New` and `CATH`.

| Dataset | EC numbers | Number of Proteins |
|---------|------------|--------------------|
| Price   | 1.-.-.-    | 59 |
|         | 2.-.-.-    | 29 |
|         | 3.-.-.-    | 21 |
|         | 4.-.-.-    | 26 |
|         | 5.-.-.-    | 9 |
|         | 6.-.-.-    | 8 |
| New     | 1.-.-.-    | 103 |
|         | 2.-.-.-    | 120 |
|         | 3.-.-.-    | 95 |
|         | 4.-.-.-    | 165 |
|         | 5.-.-.-    | 8 |
|         | 6.-.-.-    | 8 |
|         | 7.-.-.-    | 4 |
| CATH    | 2.4.1.129  | 17 |
|         | 3.4.16.4   | 35 |
|         | 3.5.1.2    | 54 |
|         | 3.5.2.6    | 9 |

## A2.3 IMPLEMENTATION OF BASELINE METHODS

BioTranslator (Xu et al., 2023) is a zero-shot classifier that operates on protein sequences and utilizes natural language, such as biological descriptions of proteins, as inputs. To determine the likelihood that a specific protein function corresponds to an EC number, we feed the text "EC X.X.X.X" along with the protein sequence into our model and extract the resulting probability. Subsequently, we employ a conventional multi-label classification approach using a threshold of $0.5$ to compute the performance metrics.

## A2.4 UNSEEN FUNCTIONALITY

We obtain 6 protein (Accession: Q20086, Q22181, P34317, Q21706, Q8GWB7, Q22295) with an unseen EC number 2.4.1.17 during training. These sequences exhibit dissimilarity in multiple facets including : (1) the lengths are different, where the shortest sequence has 507 residues and the longest sequence has 537 residues; (2) the genes are different: Q20086 has a gene annotation of "ugt-58" while the Q22181 has "ugt-47".

## A2.5 CATH SUB-SAMPLING

To derive a subset of protein sequences from the massive `CATH` dataset, we narrow our attention to the superfamily 3.40.710.10 (DD-peptidase/beta-lactamase superfamily), comprising 2194 domains. Next, we select the initial 100 functional families (FunFams) and retrieve the proteins belonging to these families. Subsequently, we eliminate duplicate sequences and ultimately acquire a set of 99 samples, each containing distinct sequences that do not overlap with those in the train set.

## A3 MORE EXPERIMENTAL RESULTS

### A3.1 PEEP FOR PROTEIN-LIGAND BINDING PREDICTION.

The classifier we have installed in PEEP can be used to distinguish between the "negative" (*i.e.*, the fused representation of a protein and a mismatched ligand) and the "positive" representation (*i.e.*, the fused representation of a protein and a matched ligand). We conduct experiments on the `Price` dataset. The trained classifier in PEEP significantly out-performs the baseline that randomly determines the protein-ligand binding relationship.

Table A9: Prediction performance of whether the given ligands are the cognate ligands of the given protein. The experiments are carried out on `Price`.

| Methods | Recall | Precision | F-1 |
|---|---|---|---|
| Random | 0.467 | 0.034 | 0.061 |
| PEEP | 0.780 | 0.184 | 0.270 |

### A3.2 ADDITIONAL ANALYSIS

**Fuse vectors of zeros as ligand embeddings during inference.** We compare two different strategies for representation fusion at inference: ① *Fuse Zero Vectors*, where we input zero vectors as ligand embeddings to calculate the representation of functional annotation and that of protein sequences with unknown functionality; and ② *Fuse Every*, we merge the protein representations from the training set with their corresponding ligand representations to compute the representation of functionalities. Furthermore, during the computation of the distance between the input protein representation and a representation of functionality, we integrate it with the representations of ligands associated with that functionality.

Table A10 demonstrates that the "Fuse Every" mode integrates ligand embeddings during inference. However, utilizing a zero-filled vector as the ligand embedding yields even more superior results in annotating previously unobserved proteins. This plausibility stems from the fact that ligand representations primarily serve as supplementary information, rendering them less informative in comparison to sequence data. Additionally, the utilization of certain ligands in multiple reactions may introduce confusion to the model. Nonetheless, incorporating ligand representations remains advantageous for the training process manifested by the improved performance compared to baselines.

Table A10: Ablation studies for different fusion strategies for inference in our proposed `PEEP` on the `Price` dataset.

| Strategy | Rec. | Prec. | F-1 |
|---|---|---|---|
| Fuse Zero Vectors | 0.211 | 0.319 | 0.240 |
| Fuse Every | 0.171 | 0.251 | 0.196 |

**Evaluating `PEEP` across Different Benchmarks.** We persist in our assessment of PEEP on the sampled subset of CATH. We first train PEEP on the training set with $10\%$ sequence identity, and then utilize the trained model to generate annotations for the proteins in the subset. PEEP attains a {recall, precision, F-1} score of {0.139, 0.062, 0.086}, surpassing CLEAN which achieves a {recall, precision, F-1} score of {0.070, 0.040, 0.049}.

**The number of ligands associated with EC numbers.** We display a histogram depicting the distribution of ligands associated with various EC numbers in Figure A6.

**Numerical results on active sites detection.** We have established another evaluation protocol to judge the performance of active sites detection, using the accuracy and the precision at residue levels. Our method achieves an average accuracy of $48\%$ and an average precision of $73\%$ on the New-392 dataset. We have also evaluated ProteInfer (Sanderson et al., 2023), using the class activation mapping (CAM) technique to identify functional localisation, on New-392, which shows an average accuracy of $49\%$ and an average precision of $54\%$. Note that both methods are trained on the same split of data ($10\%$ threshold), and the comparisons are conducted on test samples that are in the training set, a prerequisite required by ProteInfer. Our results exhibits much higher precision while having the same level of accuracy.

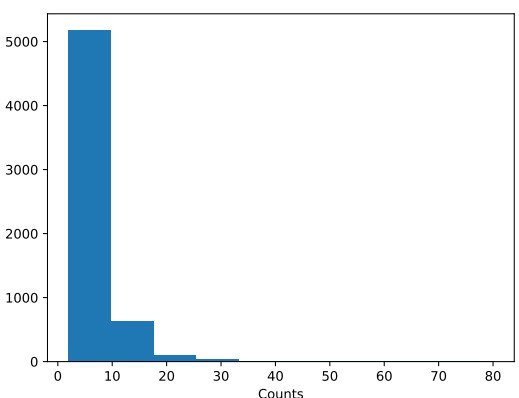

Figure A6: The histogram of number of ligands associated with EC numbers.

