# OpenReview forum: "($\texttt{PEEP}$) $\textbf{P}$redicting $\textbf{E}$nzym$\textbf{e}$ $\textbf{P}$romiscuity with its Molecule Mate – an Attentive Metric Learning Solution"
_ICLR.cc/2024/Conference — Submitted to ICLR 2024_

### Official Review · Reviewer_QFnk · 2023-10-26

**Soundness:** 2 fair
**Presentation:** 3 good
**Contribution:** 2 fair
**Rating:** 5
**Confidence:** 2

**Summary:**

This paper proposes a novel ML algorithm, PEEP, to predict enzyme promiscuity, which integrates biology priors of protein functionality to regularize the model learning.

**Strengths:**

1) This work introduces PEEP, a metric learning method, designed to identify promiscuous enzymes that can be utilized in subsequent protein engineering processes.

2)  OThe PEEP framework integrates biology-aware designs to enhance the learning of protein functionality. (1) the integration of cognate ligands' embeddings ; (2) the incorporation of an attentive module to identify crucial residues for protein functionality; and (3) the introduction of an EC-aware training objective to bolster the metric learning capability.

3) code and dataset are available.

**Weaknesses:**

1. The technique contribution of this work is not very high since it mainly use MoCo in the protein domain.It also mixes up some biological efforts to improve performance.

2. In the experiments, the dataset may be very small. e.g., very small number of proteins.

**Questions:**

1. The technique contribution of this work is not very high since it mainly use MoCo in the protein domain.It also mixes up some biological efforts to improve performance.

---

> ### Author Response · Authors · 2023-11-21
> **Response to Reviewer QFnk**
>
> We sincerely thank the reviewer for providing the valuable feedback. We have provided responses to your questions and we hope that they can address your concerns.
>
> **Weakness 1: Technique Contributions?**
> We respectfully disagree. While we do utilize MoCo in the protein domain, our key innovations lie in integrating biological insights into various facets of our approach. These aspects are not explicitly considered in previous machine learning algorithms for protein functional annotation. In our framework, we have leveraged the attention mechanism to capture residue importance, introduced a hierarchical loss term customized for functional annotations, and integrated ligand embeddings to enrich the representations. These elements collectively contribute to our framework's efficacy. Our detailed experimental results (summarized below) underscore that these incorporations, rather than using MoCo alone, are instrumental in achieving the performance enhancements we have observed.
>
> | Method | Rec. | Prec. | F-1 |
> | :-: | :-: | :-: | :-: |
> | CLEAN | 0.187 | 0.261 | 0.204 |
> | CLEAN + MoCo | 0.186 | 0.271 | 0.202 |
> | Ours | 0.217 | 0.339 | 0.241 |
>
> **Weakness 2: Sizes of datasets?**
> We would like to clarify that the datasets utilized in our experiments are not characterized as small. Specifically, for the training sets, we worked with a substantial number of protein sequences, with counts of 8K/10K/220K when employing different identity cutoffs of 10%/30%/100%, respectively. As for the testing set, which includes the Price, New, and CATH subsets, the respective sequence counts are 149/392/98. It is important to highlight that these dataset sizes align with those commonly encountered in the existing literature and are considered challenging, as demonstrated in the recent study by Yu et al. (2023).

---

> ### Author Response · Authors · 2023-11-23
>
> Dear Reviewer QFnk,
>
> We thank Reviewer QFnk for spending time to review our work and providing valuable feedbacks. We have provided responses to your question regarding the technical contribution. As the rebuttal deadline is drawing near, we hope you could take a look at our responses and see if they have addressed your concerns. Thank you again for your time and effort.
>
> Best,
>
> Authors

---

### Official Review · Reviewer_nNj8 · 2023-10-31

**Soundness:** 3 good
**Presentation:** 3 good
**Contribution:** 2 fair
**Rating:** 6
**Confidence:** 3

**Summary:**

This paper focuses on a specific bioinformatic problem, named function annotation of proteins. Previous studies like CLEAN have achieved good performance, yet they also suffer from generation ability. To achieve this problem, this paper proposes a novel metric learning method. First, at the input level, SMILE representation is used to preserve the prior of proteins. At the objective, they propose a
metric learning objective that captures the hierarchical nature of EC numbers to appropriately weight
dissimilarity at each EC level. Three datasets and extensive experiments have varified the effectiveness of the proposed method.

**Strengths:**

1. The proposed metric-based method seems simple but effective, the fusion of prior information in function annotation is an interesting idea.

2. The strong performance when given a few training data and strong generalization ability.

3. The detailed ablation experiments verified the modules on the Price dataset.

**Weaknesses:**

1. The paper might lack key explanations, such as references to EC in the second paragraph.

2. The differences between the CLEAN method and the proposed method need further discussion. I did not see a concrete motivation for CLEAN.

**Questions:**

How to use the transformer layer to achieve "facilitate learning functional residues associated with an enzyme function"?

---

> ### Author Response · Authors · 2023-11-21
> **Response to Reviewer nNj8**
>
> We genuinely thank the reviewer for saying that our idea is interesting. We have provided answers to your questions in detail.
>
> **Weakness 1: Lacking explanation?**
> Thank you for your valuable feedback. In the manuscript, we have included a paragraph explaining the EC number in Section 3.1. In response to your suggestion, we have also incorporated an introductory mention of the EC number in the second paragraph of the revised draft to provide additional context.
>
> **Weakness 2: Difference between CLEAN and PEEP.**
> The primary disparity between our approach and CLEAN lies in our incorporation of several techniques that account for biological priors across various dimensions: (1) at the input level, we fuse the representation of ligands to provide substrate-scope information; (2) at the model level, we adopt attention mechanism to learn functional residues; (3) at the objective level, we propose a metric learning objective to capture hierarchical nature of EC numbers. These enhancements collectively set our method apart from CLEAN and underline the motivations behind our approach.
>
>
> **Question 1: How to use attention modules to learn functional residues?**
> The transformer layer was chosen to replace global average pooling used by CLEAN based on the insight that the residues responsible for endowing an enzyme function make up a small amount of the sequence, thus, global average pooling will significantly weaken this signal. By using attention, we instead learn a weighted pooling that weights residues, and their importance, towards enzyme function.
>
> In Figure 2 and 4, we have provided evidence of the concordance between these attention values and the predicted significance of residues. As such, these introduced modules play a crucial role in enhancing the learning process for functional residues.

---

### Official Review · Reviewer_sx29 · 2023-11-02

**Soundness:** 3 good
**Presentation:** 3 good
**Contribution:** 3 good
**Rating:** 5
**Confidence:** 4

**Summary:**

The authors propose a ML algorithm, PEEP, to predict the function of enzymes.  The method integrates three different aspects into its pipeline: 1) It utilizes the EC number hierarchy to define different levels of similarity. 2) It uses self-attention to capture residues at the active pockets in binding to ligands. 3) It fuses the information of a protein’s ligands (their SMILES representation) with its own sequence representation by traversing the substrates and the products involved in different reactions. The method is validated on three public benchmarks and shows performance improvements in F-1 scores. It also generalizes to unseen protein sequences with unseen functionalities.

**Strengths:**

The authors demonstrate deep domain knowledge to find three different aspects in protein structure prediction that can be added to improve performance.

Detailed experimental studies have been carried out with all different kinds of optimizations, attention mechanisms, and regularizations.

Ablation studies demonstrate that all three factors are needed to improve prediction. However, it is interesting that the performance becomes worse by adding EC metric netween the 4th and 5th rows of Table 2.

**Weaknesses:**

The ideas do not seem to be that novel. The methods make use of the available data in interesting ways but there is limited ML innovation.

It is difficult to understand the significance of the performance improvement, especially when all the values are so low (e.g., on the CATH dataset).

**Questions:**

Please refer to the weaknesses.

---

> ### Author Response · Authors · 2023-11-21
> **Response to Reviewer sx29**
>
> We thank the reviewer for acknowledging the domain knowledge embedded in our designs. We have provided answers to your questions below in detail and hope they can address your concerns.
>
> **Weakness 1: Novelty of our framework?**
> While some of our techniques resemble some prevalent ones, it is also important to note that our proposed method has been recognized as novel by Reviewer nNj8 and QFnk. The novelty of our method primarily stems from its design, which is driven by specific challenges in the biological domain. This includes three key innovations: (1) the functions of proteins are highly correlated with their active sites; (2) the functional annotations have hierarchical structures; and (3) embeddings of ligands can be leveraged as additional sources of information for functional annotations. These elements, not explicitly considered in prior work, are central to our innovative approach. We have incorporated an attention mechanism to capture residue importance, introduced a hierarchical loss term customized for functional annotations, and integrated ligand embeddings to enrich the representations. Empirical evidence supports the effectiveness of these integrations in enhancing functional annotation performance.
>
> **Weakness 2: Significance of the performance improvement?**
> We performed an additional analysis to assess the significance of the performance improvements on the Price-149 and CATH datasets, both of which were subjected to three separate experiments under a 10% sequence identity cutoff. Our results revealed error bars of approximately 0.007 and 0.011 for Price-149 and CATH, respectively, demonstrating the statistical significance of the observed improvements.
>
> **Weakness 3: Performance Comparison in Table 2.**
> Thank you for your observation! It is worth noting that when comparing the results between the 4th and 5th rows of Table 2, we observe improvements in four metrics out of six. As such, we continue to believe that the combination of all techniques remains the most favorable choice.

---

> > ### Comment · Reviewer_sx29 · 2023-11-22
> > **Thanks for your response**
> >
> > I thank the authors for their response.
> > I will maintain my score.

---

### Official Review · Reviewer_siYE · 2023-11-09

**Soundness:** 3 good
**Presentation:** 4 excellent
**Contribution:** 3 good
**Rating:** 6
**Confidence:** 4

**Summary:**

The authors focus on the problem of predicting an enzyme's function given it's amino acid sequence, with a focus on situations where the sequence is quite remote from any enzyme with known function or where we seek to make predictions for a novel enzyme function that had not previously appeared in the training data. They use a number of techniques for improving models, such as triple-loss-based contrastive learning, using pre-trained protein and small molecule embeddings, a loss function that accounts for the label hierarchy, and an attention-based pooling technique.

**Strengths:**

The intro/background is accessible, comprehensive, and well written.

The problem that the paper approaches is important for the life science community and the paper achieves reasonable performance improvements.

The authors are careful to construct train-test splits that probe a model's ability to do meaningful extrapolation, both in terms of protein sequence and protein function.

The paper provides a significant number of ablations probing the impact of various design choices.

**Weaknesses:**

Despite the paper's title, it doesn't really model enzyme promiscuity, just enzyme function. Promiscuity is the tendency of an enzyme to accept many substrates. There is no evaluation setup that focuses on enzymes with annotated activity on multiple substrates. In part this is because Swissprot, the training data, only contains annotations for enzymes' natural function, while there are many other reactions that these enzymes could catalyze in the lab.

The paper boasts about incorporating 'biology priors', but the actual solutions aren't particularly novel or biology-specific. For example, the authors use a learned attention head to pool per-residue embeddings into a per-sequence embedding. The motivation about attention as focusing on active site residues is tenuous and post-hoc. Attention-based pooling is common these days. Similarly, another 'biology' detail is that the labels are hierarchical in nature, and the loss function is adjusted such that the similarity of 2 proteins' embeddings reflects the degree of their similarity in terms of the hierarchy of function. There's nothing biology-specific about training models with hierarchical labels.

It's unclear if the benchmarking setup is fair to baseline models. See below.

**Questions:**

I'm confused by the comparison to baseline models. When you compare to proteinfer, for example, do you retrain it on the same train-test split as your model? If not, how is the comparison fair?

I'm confused by the motivation for changing the clustering threshold for de-duplicating the training data. To me, the key quantity when assessing the difficulty of an extrapolation task is the distance between evaluation examples and training examples. How does this distance change as you vary the training threshold?

Can you elaborate on the concept of 'promiscuity'? In what sense are you tackling the promiscuity problem?

I was confused by the statement in the intro that 'only 570K (∼ 0.3%) sequences have been manually annotated with computational methods that bridge the sequence-annotation gap.' Surely there are far more sequences in uniprot with computationally-derived functional annotations. Do you mean that there are 570K sequences with human-curated annotations (i.e., Swissprot)?

I found fig 4 unsatisfactory, since the distributions seem to overlap so much. Is there a way to quantitatively evaluate this separation, such as a precision/recall/f1 for classifying residues as occurring in the active site?

==Update after authors' response==
Thank you for clarifying so many of my questions. Many of my concerns regarding technical soundness have been addressed. However, I continue to feel that the methodological novelty of the paper is lower than many ICLR papers. Of course, there are accepted papers that have low novelty, but they typically introduce new tasks, datasets, or provide interesting extensions of methods.  I have raised my score to 'weak accept', but acknowledge that the paper is truly borderline and unfortunately may not get accepted due to the space constraints at ICLR.

---

> ### Author Response · Authors · 2023-11-21
> **Response to Reviewer siYE**
>
> We appreciate the efforts made by the reviewer for providing the detailed and valuable comments. We have provided responses in detail and we hope they can address your concerns.
>
> **Weakness 1: Concept of Promiscuity**
> We appreciate and agree with your comments. Your understanding of enzyme promiscuity is correct. While we have proteins with multiple EC numbers in our testing datasets, we are not directly tackling the promiscuity problem. We have updated the title of the paper and also the draft to make it purely about enzyme function and not enzyme promiscuity.
>
> **Weakness 2: Motivation of using attention?**
> We would like to clarify that our motivation for deploying attention mechanisms is based on specific scientific reasoning and not post-hoc. In the context of protein analysis, it is well-established that not all residues contribute equally to the protein's function. Traditional methods that assign uniform weights to all residues often overlook the nuanced contributions of key functional residues.
> Our choice of attention mechanisms is driven by their ability to dynamically emphasize these functionally significant residues. This is not merely following a common trend, but a strategy to enhance model accuracy. These attention modules allow our model to adaptively focus on residues that are crucial for the protein's activity, thereby offering a more precise representation of functional sites compared to uniform weighting.
> Empirical evidence supports the effectiveness of attention mechanisms. In our experiments, deploying the attention mechanism consistently demonstrated superior performance. This improvement is attributable to the model's ability to learn and prioritize residues that are more relevant to the protein's function.
> Thus, our adoption of attention mechanisms is a design that aims at addressing the specific challenges in protein function prediction.
>
>
> **Weakness 3: Hierarchical label training is not biology-specific**
> We appreciate the reviewer's point that hierarchical labels are not exclusively used in biological contexts. However, we demonstrate that EC numbers’s ontology is well-suited for the hierarchical label strategy and we demonstrate that it leads to improved performance. Thus, demonstrating the use of domain knowledge via hierarchical labeling is beneficial in prediction protein function from sequence. An insight that could benefit the protein machine learning community.
>
> Here are a few examples on how combining hierarchical labels with domain knowledge is beneficial:
> - For example, [r1] incorporates class hierarchy from WordNet [r2] as a source of domain knowledge to improve the performance of visual classification and [r3] uses label hierarchy to improve concept classification. Our technique applies this concept to the domain of functional annotation. By adjusting the loss function to reflect protein similarities based on their functional hierarchy, our model effectively integrates domain-specific knowledge.
>
> In summary, while hierarchical labeling is a technique used across various fields, its implementation in our model is specifically tailored to address the complexities of biological data. This approach aligns with established practices in the literature, demonstrating our model's effective absorption and utilization of domain-specific knowledge.
>
>
> [r1] Integrating Domain Knowledge: Using Hierarchies to Improve Deep Classifiers
>
> [r2] WordNet: A Lexical Database for English
>
> [r3] Domain-Specific Knowledge Acquisition and Classification using WordNet

---

> ### Author Response · Authors · 2023-11-21
> **Response to Reviewer siYE (cont'd)**
>
> **Question 1: The fairness of comparison with baselines.**
> In our study, we have ensured a fair comparison with baseline methods. For the training phase, we have used identical training datasets to (re-)train all models. For the evaluation, all the models including baselines, are evaluated on identical sets of test data. This ensures the fairness of our comparison. We have incorporated this benchmarking setup description into the draft to enhance its clarity and presentation.
>
> **Question 2: Motivation for de-duplicating training data.**
> The motivation of using different clustering thresholds is to decrease the similarity between the training and the testing set and evaluate the model’s ability to generalize. This is in line with your understanding on the importance of the difference between training and testing examples in changing the difficulty of extrapolation tasks.
> In response to your inquiries, we conducted an assessment of relative Levenshtein distances (i.e., divided by the length of the sequence) between sequences in the training and testing sets. For every sequence in the testing set, we average the distance between it and 5 samples from the training set with the smallest Levenshtein distances. The table presented below illustrates that as the thresholds decrease, the distances between these sequences tend to increase. This indicates that adjusting the clustering threshold effectively enhances the distinctiveness of the training and testing data.
>
> | Clustering Threshold | Average Relative Levenshtein Distance |
> | :-: | :-: |
> | 100% | 0.615 |
> | 30% | 0.713 |
> | 10% | 0.721 |
>
>
> **Question 3: Clarifying the statement in introduction.**
> Thank you for pointing out this ambiguity in our statement. Yes, this number refers to the number of sequences with experimental annotation in UniProt SwissProt. We will clarify that this refers to SwissProt in the main text.
>
> **Question 4: Quantitative evaluation of active sites detection.**
> Following your suggestion, we have established another evaluation protocol to judge the performance of active sites detection, using the accuracy and the precision at residue levels. Our method achieves an average accuracy of 48% and an average precision of 73% on the New-392 dataset. We have also benchmarked ProteInfer, using the class activation mapping (CAM) technique to identify functional localization, on New-392, which shows an average accuracy of 49% and an average precision of 54%. Note that both methods are trained on the same split of data (10% identity), and the comparisons are conducted on test samples that are in the training set, a prerequisite required by ProteInfer. Our results exhibit much higher precision while having the same level of accuracy. We have included these new results in the revision.

---

> ### Author Response · Authors · 2023-11-23
>
> Dear Reviewer siYE,
>
> We extend our gratitude to Reviewer siYE for dedicating time to review our work and for the valuable constructive comments provided. We have meticulously addressed your questions in a point-by-point fashion. As the rebuttal deadline is drawing near, we hope you could take a look at our responses and see if they have addressed your concerns. Thank you again for your time and effort.
>
> Best,
>
> Authors

---

### Official Review · Reviewer_fk1t · 2023-11-09

**Soundness:** 3 good
**Presentation:** 3 good
**Contribution:** 2 fair
**Rating:** 6
**Confidence:** 3

**Summary:**

The paper proposes PEEP, a metric learning framework for protein functionality. The framework uses established techniques, such as ESM2 embeddings and Momentum Contrast as the backbone of the model. On top of these, the paper applies three types of algorithmic insights: using the ligands as additional information, leveraging a self-attentive mechanism to identify key residues, and a modified objective focused on the hierarchy of EC labels. PEEP outperforms relevant baselines on Price-149, New-392 and CATH and ablation studies show the relevance of the proposed changes.

**Strengths:**

The paper is well written and motivated. The final architecture is backed by good empirical performance, and ablation studies are thorough, providing necessary insight into the importance of the proposed modifications.

**Weaknesses:**

I encourage the authors to revise the writing in "To meet the goal, we customize a self-attention mechanism (Figure 1, c) to model the residue importance within protein sequences” as it currently reads as though the self-attention mechanism is newly-proposed, while it seems to be a standard setup.

While the paper is well-motivated, it seems that a significant part of it studies how some well-known techniques fit together in the context of the chosen task. The lack of novelty hinders from a higher rating at the moment unfortunately.

**Questions:**

In the paper, it is stated that "PEEP randomly samples one from the ligands’ SMILE embedding and integrates it with the protein’s sequence representation”. It would be useful to provide additional information on the distribution of ligands per protein and the sensitivity of the random choice with respect to the results.

Moreover, in 3.3, two methods are described for doing the fusion — how does the performance compare/which one is used?

---

> ### Author Response · Authors · 2023-11-21
> **Response to Reviewer fk1t**
>
> We greatly appreciate the reviewer's recognition of the quality of our paper's writing and its empirical performance. We have prepared detailed responses to your comments, and we hope that they can address your concerns.
>
> **Weakness 1: revision of the writing?**
> Thank you for your insightful feedback. We have revised the manuscript and adjusted the sentence to enhance clarity, addressing your suggestion as follows: “To meet the goal, we customize a self-attention mechanism (Figure 1, c) to model the residue importance within protein sequences,” has been modified to “To meet the goal, we deploy a self-attention mechanism (Figure 1, c) to model the residue importance within protein sequences,”.
>
> **Weakness 2: lack of novelty?**
> While some of our techniques resemble some prevalent ones, it is also important to note that our proposed method has been recognized as novel by Reviewer nNj8 and QFnk. The novelty of our method primarily stems from its design, which is driven by specific challenges in the biological domain, including three key aspects: (1) the functions of proteins are highly correlated with their active sites; (2) the functional annotations have hierarchical structures; and (3) embeddings of ligands can be leveraged as additional sources of information for functional annotations. These elements, not explicitly considered in prior work, are central to our innovative approach. We have incorporated an attention mechanism to capture residue importance, introduced a hierarchical loss term customized for functional annotations, and integrated ligand embeddings to enrich the representations. Empirical evidence supports the effectiveness of these integrations in enhancing functional annotation performance.
>
> **Question 1: more information about ligands?**
> Thank you for your feedback. To address your concern, we have included a histogram illustrating the distribution of ligands associated with proteins in our training dataset in Appendix. We can see most EC numbers have less than 10 associated ligands.
>
> We would like to emphasize that our random selection process is performed iteratively at each training step, ensuring that all ligands are utilized in the training procedure. We have also conducted multiple experiments specifically on the Price-149 dataset, employing a sequence identity cut-off of 10%. These experiments reveal a negligible error bar of 0.007, underscoring that the random selection process introduces only minor fluctuations in the results.
>
> **Question 2: methods for fusion?**
> We have chosen the second option, which involves fusing protein representations with zero vectors. For a detailed performance comparison between the two methods, you can refer to Appendix A3.2. To make it more convenient, we have included the results in the table below:
>
> | Strategy | Rec. | Prec. | F-1 |
> | :------: | :-----: | :-----: | :--: |
> | Fuse Zero Vectors | 0.211 | 0.319 | 0.240 |
> | Fuse Every | 0.171 | 0.251 | 0.196 |

---

> > ### Comment · Reviewer_fk1t · 2023-11-23
> >
> > I would like to thank the authors for their response. I will maintain my score, recommending acceptance of the paper.

---

### Author Response · Authors · 2023-11-21
**General Responses**

We would like to thank all the reviewers for their efforts and valuable comments on our work. In the rebuttal period, we have provided point-by-point responses to all the comments and questions.

In the revision of our work, we have:
- Changed the title and also change the word ‘promiscuity’ to improve the presentation (@Reviewer siYE)
- Added a sentence to describe EC number (@Reviewer nNj8), and also clarified the statement regarding the number of functional annotations (@Reviewer siYE)
- Added experiments to quantitatively evaluate the performance of active sites detection (@Reviewer siYE)

We would appreciate all the reviewer comments again. Should there be any further questions or clarifications needed, we are willing to provide additional information.

---

### Meta-Review · Area_Chair_bQDj · 2023-12-06

**Metareview:**

The submission proposes an ML algorithm (PEEP) to predict protein function, combining several components (biologically inspired regularizer, self-attention, protein function label hierarchy based metric learning objective), and demonstrate its superior performance on three public benchmarks.

Reviewers agree that the submission is borderline. Several concerns were raised around the empirical evaluation, but after discussion/revision have been mostly addressed. Still, the submission provides limited ML novelty and the application domain and task are also not novel. The main contribution of the work is in improved performance on the specific biological applications, which may have limited interest to the broader ICLR community, and so I do not recommend publishing in it's current form.

It may help if a future submission of this work placed the importance of the empirical results or application domain in a context that the broader ICLR community can appreciate and/or clearly demonstrated novel general ML methodology contributions.

**Justification For Why Not Higher Score:**

There is not much ML novelty in the submission and the application itself is not new or ground-breaking. In my opinion, this submission would better serve the audience of a biology (protein) focused journal/conference.

**Justification For Why Not Lower Score:**

N/A

---

### Decision · Program_Chairs · 2024-01-16

Reject